# Effect of Post-Printing Cooling Conditions on the Properties of ULTEM Printed Parts

**DOI:** 10.3390/polym15020324

**Published:** 2023-01-08

**Authors:** Tatjana Glaskova-Kuzmina, Didzis Dejus, Jānis Jātnieks, Andrey Aniskevich, Jevgenijs Sevcenko, Anatolijs Sarakovskis, Aleksejs Zolotarjovs

**Affiliations:** 1Baltic3D.eu, Braslas 22D, LV-1035 Riga, Latvia; 2Institute for Mechanics of Materials, University of Latvia, Jelgavas 3, LV-1004 Riga, Latvia; 3Institute of Solid State Physics, Kengaraga 8, LV-1063 Riga, Latvia

**Keywords:** ULTEM, fused deposition modelling, thermal history, cooling conditions, mechanical properties, thermophysical properties

## Abstract

This paper aimed to estimate the effect of post-printing cooling conditions on the tensile and thermophysical properties of ULTEM^®^ 9085 printed parts processed by fused deposition modeling (FDM). Three different cooling conditions were applied after printing Ultem samples: from 180 °C to room temperature (RT) for 4 h in the printer (P), rapid removal from the printer and cooling from 200 °C to RT for 4 h in the oven (O), and cooling at RT (R). Tensile tests and dynamic mechanical thermal analysis (DMTA) were carried out on samples printed in three orthogonal planes to investigate the effect of the post-printing cooling conditions on their mechanical and thermophysical properties. Optical microscopy was employed to relate the corresponding macrostructure to the mechanical performance of the material. The results obtained showed almost no difference between samples cooled either in the printer or oven and a notable difference for samples cooled at room temperature. Moreover, the lowest mechanical performance and sensitivity to the thermal cooling conditions were defined for the Z printing direction due to anisotropic nature of FDM and debonding among layers.

## 1. Introduction

Additive manufacturing (AM) of both single- and multi-material structures has recently been applied to efficiently produce complex structures, thus saving production time and resources. AM offers unprecedented levels of freedom for the design and application of 3D-printed polymer materials, e.g., in automotive, aerospace, biomedical, and dentistry fields [1]. Currently, fused deposition modeling (FDM) is one of the AM technologies that has been extensively applied in the manufacture of 3D-printed polymer parts [2,3]. During this process, a polymer is extruded through a heated nozzle and deposited in a semi-molten state to create the required shape via sequential build-up of layered depositions [4].

One of the main disadvantages of FDM-processed polymeric parts is their highly anisotropic nature due to the intrinsic properties of the extruded filament, oriented build process, and limited degree of fusion between the layers [5,6]. These issues lead to anisotropy in the mechanical and thermophysical properties of 3D-printed polymer parts, especially in the direction perpendicular to the construction layer (XZ) [2,4,5,6,7]. Furthermore, due to voids and porosity that are inherent to the FDM printing process, polymer samples manufactured by FDM typically have lower mechanical strength and possess lower apparent mechanical properties compared to parts manufactured using traditional processing techniques, such as compression and injection moldings [8].

Furthermore, it is well known that the properties of 3D-printed parts processed by FDM strongly depend on the building process parameters (e.g., infill percentage, build and raster orientation, layer thickness and width, feed rate, printing speed, nozzle diameter, etc.). The effect of the building process parameters on the mechanical (tensile, flexural, impact) [9,10,11,12,13,14,15], flame-retardant [2,12], and thermal properties [16,17] of 3D-printed parts has been thoroughly discussed in the literature.

Additionally, 3D-printed polymer parts exhibit high sensitivity to thermal processing conditions [8,18,19,20,21,22]. Both the temperature of the heating element (model build temperature) and temperature around the printed part (envelope temperature) have been shown to affect the quality and mechanical performance of the printed parts [18]. The results of fracture analysis also confirmed that an increase in the envelope temperature positively contributes to neck (bond formation area) development between the layers [19].

An increase in the printing bed temperature above the filament’s glass transition temperature was attributed to better adhesion between the printing bed and filament [18]. Considering inhomogeneity in the convective air flow field in the printer, the position of the printed part on the build plate can result in different thermal histories, which can affect the development of interlayer strength [20]. Moreover, the incorporation of ultrasonic vibrations [7] and laser pre-deposition heating [23] in FDM have resulted in better mechanical properties of 3D-printed parts.

For use in transportation and aerospace applications, ULTEM 9085^®^ (Ultem), an amorphous thermoplastic material, is particularly distinguished among similar materials for its flame, smoke, and toxicity certification [24,25]. The effects of thermal annealing on the glass transition temperature of FDM-processed Ultem parts were analyzed and found to be crucial for both their flexural [21] and tensile [22] properties.

A robotic system fully controlled by automatic software was designed and developed at AM Craft (Riga, Latvia) to produce 3D-printed parts day and night (Appendix A). Fast removal from the printer and cooling in an oven with a similar thermal cooling profile as the printer could allow printing of more samples, thus improving the productivity of the 3D printing process.

However, rapid cooling of the printed parts can lead to inhomogeneous temperature distribution in the printed parts and, as a result, in volumetric shrinkage that generates residual stresses, pre-existing cracks, and preliminary failure of the part during service [21]. Therefore, the effect of thermal (cooling) conditions on the mechanical properties of 3D-printed polymer parts should be studied thoroughly. Three different cooling conditions were applied to Ultem samples: in the printer (P), rapid removal from the printer and cooling under similar thermal conditions in the oven (O), and cooling at room temperature conditions (R). Thus, for samples having different post-printing cooling histories, equality/similarity of mechanical and thermophysical properties could be a scientifically based justification that the different thermal histories of the printed parts will not have significant consequences on these properties, and such cooling conditions could be applied without risk to the products’ quality and performance.

This work aimed to estimate the effect of post-printing cooling conditions on the tensile and thermophysical properties of Ultem processed by FDM. Further results on fatigue testing at different stress levels (75%, 50%, and 25% of the ultimate strength) will be summarized and published in the following paper. Hopefully, these results will positively contribute to improving the productivity and efficiency of 3D printer machines.

## 2. Materials and Methods

### 2.1. Materials and Manufacturing of the Test Samples

The investigated material was ULTEM^®^ 9085, which is a blend of polyetherimide and polycarbonate, provided by Stratasys (Eden Prairie, MN, USA). It was used at Baltic3D.eu (Riga, Latvia) to produce samples for tensile and dynamic mechanical analysis (DMTA) from the same batch. The test samples were printed using a Stratasys F900 machine (Eden Prairie, MN, USA) in directions X, Y, and Z, as reported in ref. [2] and shown in Figure 1.

The printing parameters of all samples are provided in Table 1. Apart from the variation in printing direction, the rest of the printing parameters were kept the same. The infill density was set to 100% (solid) for all samples and the samples were printed without a border. The dimensions of the test specimens for the tensile tests were 100 × 10 × 3 mm^3^ according to the standard [26]. Five samples for each printing direction and cooling regime were manufactured. The dimensions of the samples for dynamic mechanical testing were 30 × 3 × 1 mm^3^. Three replicants were tested to obtain statistically confident values. Thus, the values provided on the graphs correspond to the mean value together with the standard deviation.

### 2.2. Methods

#### 2.2.1. Cooling of the Test Samples

Immediately after printing, the samples were subjected to three different cooling conditions: cooling in the printer from 180 to 45 °C for 4 h (P), rapid removal from the printer and cooling in the oven from 200 to 45 °C during 4 h (O), and removal from the printer and cooling at room temperature (R). The internal dimensions of the printer and oven were 1.6 × 1.2 × 1 m^3^ and 1.14 × 0.8 × 0.94 m^3^, respectfully. The kinetics of the sample cooling within these regimes were registered using a digital multimeter Düwi 07975 (Bieruń, Poland) with a thermocouple. Thus, the samples were subjected to free cooling in the printer and at room temperature, while cooling in the oven was simulated according to printer’s thermal conditions. The samples were stored in these conditions until they reached room temperature.

The thermocouple was embedded in a freshly printed dog-bone sample with a cylindrical hole of a diameter of approx. 2 mm. The temperature was recorded every minute for 4 h. The temperatures inside the printer and oven were recorded near the sample using built-in sensors. Thus, the effect of the thermal cooling conditions was established for these three sets of samples denoted as P (cooled in the printer), O (cooled in the oven), and R (cooled at RT) for samples printed in the X, Y, and Z directions. The Ultem samples during post-printing cooling in the printer, oven, and at room temperature are shown in Figure 2. The sample in the oven was in a similar position (next to the door) as in the printer.

#### 2.2.2. Morphological Analysis

The morphology of the fracture surfaces for the transverse cross-sections of the Ultem samples printed in the X, Y, and Z directions was examined using a conventional complementary metal–oxide–semiconductor (CMOS) camera with a built-in 5× zoom lens and by high-resolution SEM-FIB electron microscope (Helios 5 UX; Thermo Fisher Scientific, Waltham, MA, USA) operated at 0.5 kV and 25 pA with scan interlacing and integration to avoid charging. Optical microscopy and scanning electron microscopy (SEM) were employed to relate the corresponding macrostructure to the mechanical performance of the material along the printing directions. The specimens were used after testing their tensile properties without further modification.

#### 2.2.3. Tensile Tests

Quasi-static tensile tests were performed according to the standard described in ref. [26] for all test specimens using a Zwick 2.5 testing machine (Zwick Roell Group, Ulm, Germany) with a crosshead speed of 2 mm/min at RT until failure. Tensile strength was defined as the maximum achieved value of stress in the specimen, and the elastic modulus was calculated from the slope of a secant line between 0.05% and 0.25% strain on a stress-strain plot.

#### 2.2.4. Dynamic Mechanical Thermal Analysis

DMTA of the samples was performed using a Mettler Toledo DMA/SDTA 861 (Greifensee, Switzerland) for the evaluation of the effect of cooling conditions on the thermomechanical properties of Ultem. The testing procedure was a temperature scan from 30 to 230 °C, at a heating rate of 3 °C/min, with an applied tensile force of 0.9 N, at a frequency of 1 Hz.

## 3. Results and Discussion

### 3.1. Morphology of the Fracture Surface

The morphology of the fracture surface for transverse cross-sections of Ultem samples printed in the X, Y, and Z directions was analyzed by optical microscopy and SEM, and the results are provided in Figure 3 and Figure 4, respectfully. It should be noted that no meaningful differences were observed between samples under different thermal cooling conditions. Thus, the micrographs shown in Figure 3 and Figure 4 correspond to samples after cooling at room temperature conditions. The horizontal samples shown in Figure 3a and Figure 4a (X) and Figure 3b and Figure 4b (Y) exhibited large plastic deformations during tensile loading. Since the loading direction was the same as the direction of filament alignment, fracture occurred due to breakage of the filaments, leading to a smooth fracture surface. Moreover, both microscopic analyses revealed that the void distribution in the transverse cross-sections of samples printed in the X and Y directions was almost the same, indicating a similar degree of fiber-to-fiber fusion.

In contrast with samples printed in the X and Y directions, the examination of the morphology of samples printed in the Z direction (Figure 3c and Figure 4c) showed that the transverse cross-sections were completely different, revealing rough structures of extruded filaments [4,19]. Thus, for vertical samples (Z), mechanical failure occurred along the interlayer interface and the rough fracture surface revealed peeling and pull-out of the filaments [23]. Obviously, adhesion between the layers dominated in the fracture process, resulting in the lowest tensile strength and elastic and storage moduli observed for the samples printed in the Z direction for all cooling conditions (see Section 3.3 Tensile Properties and 3.4 Thermophysical Properties).

Similar results were obtained for the fracture behavior of samples as a function of the different building orientations [3,27]. Samples printed in the X and Y directions revealed the best tensile properties since the filament was extruded parallel to the sample axes and in the same direction as the load application. Therefore, samples printed in these two directions strongly opposed the load application. Meanwhile, the fracture surface of samples printed in the Z direction was different from that of the samples printed in the X and Y directions, indicating debonding between layers and adhesion that could not withstand high load application [1]. Thus, debonding at interfaces placed perpendicular to the tensile load was attributed to lower tensile properties (strength and elongation at break) [3,27].

### 3.2. Cooling Kinetics of the Test Samples

The temperature evolution was recorded in the printer and oven, as well as in the Ultem freshly printed dog-bone sample with an embedded thermocouple, during cooling in the printer, oven, and at RT (see Figure 2) and is shown in Figure 5. It is interesting to note that the rate of cooling of the Ultem sample was different in all three environments. Cooling was very fast at room temperature conditions and lasted only 5 min. Obviously, the cooling duration of the sample was much longer in the printer and oven, while the temperature values were higher for the same period in the oven than in the printer, indicating different cooling regimes for the oven and printer.

Since the temperatures in the printer and oven were also recorded, it was possible to compare them and describe the cooling kinetics of the sample using Newton’s law of cooling [27]:(1)$$T\left(t\right)=\left(T_{0}-T_{1}\right)\cdot e^{-rt}+T_{1}$$
where *t* is time, *T*(*t*) is the time-varying temperature of the sample, *T*_0_ is the temperature of the sample when *t* = 0, *T*_1_ is the final temperature of the environment, and *r* is a heat transfer coefficient that depends on the surface area (dimensions of the sample), material properties, and thermal cooling conditions.

According to Newton’s law of cooling, the rate of cooling or the rate of loss of heat of an object is directly proportional to the temperature difference between the object and the environment. The heat transfer coefficients depend on the cooling conditions and were evaluated to be 0.006, 0.01, and 0.4 min^−1^ for cooling in the oven, printer, and at room temperature conditions, respectfully. These values were found by the minimization of the aim function, which was the sum of the difference between the experimental data and evaluation results given by Equation (1).

As seen in Figure 5, Newton’s law of cooling described well the kinetics of the cooling processes for the printer and at room temperature conditions. Moreover, it was obvious that the temperature evolution vs. time was slower in the oven than in the printer. It should be emphasized that the oven was intentionally heated to a higher temperature (200 °C) due to the thermal loss that potentially could occur during the opening of the door when the sample was removed from the printer and put into the oven. Otherwise, the cooling conditions of the printer, oven, and samples contained within were successfully reached and the temperature values coincided after 4 h of cooling.

### 3.3. Tensile Properties

The analysis of the stress-strain curves revealed that samples subjected to cooling in different environments had almost the same stress-strain behavior, whereas the effect of the printing direction on the mechanical behavior was substantial. For samples printed in the Y direction, relatively large plastic deformation and the formation of “necking” were observed. The tensile strength, elastic modulus, and maximal deformation of samples were determined and analyzed for each printing direction and cooling condition. The results obtained are provided in Figure 6. According to the results, it should be noted that regardless of the cooling conditions, the highest tensile strength and elastic modulus corresponded to samples printed in the X and Y directions, whereas the lowest tensile strength and elastic modulus corresponded to the sample printed in the Z direction (approximately 3 times lower for tensile strength and 1.3 times lower for elastic modulus). This can be explained by the longitudinal filament orientation (along the sample length) in the case of the X and Y printing directions, which was also parallel to the tensile load application direction [1,6,28]. The transverse cross-sections analyzed by optical microscopy also revealed very similar macrostructures of the printed strands for the samples printed in the X and Y directions. However, the transverse cross-section was different for samples printed in the Z direction, displaying brittle structures of extruded filaments that could be an indication of internal defects, such as voids and uneven diameters of the strips, due to the anisotropic nature of the fused deposition modelling [4,29].

According to the material datasheet (printed with T16 tip) provided for ULTEM 9085 by Stratasys (Eden Prairie, MN, USA) [8], the tensile strength is 68.1 ± 5.7 MPa in the X and Y directions and 39.4 ± 3.1 MPa in the Z direction, while the elastic modulus is 2.52 ± 0.06 GPa in the X and Y directions and 2.41 ± 0.06 GPa in the Z direction. Therefore, it could be concluded that for the X and Y directions, the tensile strength and elastic modulus correlated well with the values of the material datasheet, but they were significantly lower for the Z direction. Similar results for the tensile strength and modulus of Ultem printed in different directions were found in the literature [1,6,29,30]. To overcome FDM-associated anisotropy leading to large differences in tensile properties, particularly in the Z-build orientation (vertical), optimal combinations of building process parameters (e.g., layer thickness, air gaps, infill percentages, feed rate, printing speeds, and raster angles) could be considered [9,31,32,33].

Moreover, Figure 6c demonstrates that the maximal deformation of the test samples was highest for the Y printing direction and samples cooled at RT revealed high values of plastic deformation. However, considering the large data scattering observed in the results, which is usually a characteristic issue for maximal deformation, the results for the oven and printer could be regarded as very close. According to the material datasheet, the maximal deformations are 5.4% (X, Y) and 1.9% (Z). All experimentally obtained results revealed much higher values (8.51 ± 1.43% for X and 2.59 ± 0.31% for Z), especially for the Y printing direction (20.73 ± 4.53%).

Finally, focusing on the effect of cooling conditions on the tensile characteristics of Ultem, one may note that cooling in the printer or oven did not have significant consequences for all printing directions and the presence of a negative effect could be attributed to data scattering within the set of specimens. Nevertheless, cooling of the specimens at RT resulted in a more significant reduction in tensile strength (by 10%) for the Y printing direction and elastic modulus (by 12–15%) for all printing directions.

### 3.4. Thermophysical Properties

Dynamic thermal mechanical analysis provides important information to identify changes in the chain mobility restriction of the polymer network, which can be analyzed regarding different cooling conditions. Figure 7 summarizes the data for Ultem studied for the storage modulus (E’) as a function of temperature for different cooling conditions: cooled in the printer (P), oven (O), and at room temperature (R). The representative curves are provided for each set of test samples. According to Figure 6, the test samples were characterized by a similar small decrease in the storage modulus in the glassy region, a sudden reduction in the glass-rubbery transition zone, and almost the same value in the rubbery region. The glass transition temperature (*T*_g_) was evaluated according to the standard described in ref. [34,35] from the storage modulus dependence on the temperature (Figure 6) at the inflection point and is shown in Figure 6 as a function of the printing direction.

Figure 8 shows that the glass transition temperature was not affected by the printing direction (X, Y, Z) and was approx. 186 ± 4 °C. Similar results were reported with minimal difference in *T*_g_ for Ultem samples printed in different directions [4,33]. Moreover, no differences were observed among curves for Ultem filaments and specimens printed in different directions obtained by thermal gravimetric analysis in both air and argon atmospheres [3]. Therefore, it was concluded that the FDM process did not affect the thermal and thermo-oxidative degradation processes in the investigated material. Additionally, the effect of the cooling conditions on the glass transition temperature was negligible.

Nevertheless, according to Figure 7, for all cooling conditions, samples printed in the Z direction were characterized by much lower values of the storage modulus in the glassy region. For samples cooled in the printer, oven, and at room temperature, the reductions were 13%, 108%, and 39%, respectfully. For printing directions X and Y, the deviation was not so prominent and could be neglected. Therefore, the most critical loads upon application should not coincide with the Z printing direction because interlayer adhesion is critical for Z-oriented parts, as under loading of the interlayer bonds in these parts will bear the applied load and as a result, Z-oriented parts possess considerably weaker mechanical properties than parts in X (flat) or Y (on-edge) orientations [3,4,7,36,37].

For further comparison of the different cooling conditions, the results obtained for the storage modulus of Ultem samples printed in the same X, Y, and Z directions are shown in Figure 9. Obviously, samples printed in the X and Y directions and cooled under different conditions were characterized by almost the same dependence of the storage modulus on the temperature. However, samples printed in the Z direction displayed a different behavior of the storage modulus as a function of temperature. Interestingly, samples cooled in the oven had the lowest storage modulus in the glassy region, which was 91% lower than that of samples cooled in the printer.

According to the material datasheet [8], the glass transition temperature of ULTEM 9085 (Stratasys, Eden Prairie, MN, USA) evaluated by differential scanning calorimetry and thermal-mechanical analysis is 177 °C. In this work, dynamic thermal mechanical analysis was used, and the glass transition temperature was evaluated as 186 ± 4 °C regardless of the cooling conditions. This value was 5–13 °C higher than that in the datasheet. Considering data scattering and the different methods employed, this deviation is acceptable.

Finally, focusing on the effect of the cooling conditions on the thermophysical characteristics of Ultem, one may note that the cooling conditions had no significant consequences for the X and Y printing directions. The storage modulus and loss factor for samples printed in these directions almost coincided with the temperature range of 30–230 °C. Nevertheless, for all cooling conditions, the Z printing direction and cooling in the oven and at room temperature resulted in a more significant reduction of the storage modulus though having almost no effect on the glass transition temperature. Therefore, applying critical loads on parts printed in this direction is not recommended since the consequences of the thermal cooling history were notable.

## 4. Conclusions

It was experimentally confirmed that the effect of post-printing cooling conditions on the tensile and thermophysical properties of ULTEM 9085 printed parts was almost the same when cooled in the printer or oven but more notable when cooled at room temperature.

The cooling of the specimens at RT resulted in a more significant reduction in the tensile strength (by 10%) and elastic modulus (by 12–15%) for all printing directions, while the results could be regarded as very close for cooling in the printer or oven, considering the observed data scattering. Due to the anisotropic nature of FDM, the tensile characteristics of Ultem 3D-printed samples were shown to significantly vary as a function of build orientation. For all cooling conditions, the highest tensile strength and elastic modulus corresponded to samples printed in the X and Y directions but were lowest for samples printed in the Z direction (approximately 3 times lower for tensile strength and 1.3 times lower for elastic modulus). Therefore, regardless of the post-printing thermal history, the most critical loads upon application should not coincide with the Z printing orientation because interlayer adhesion is critical for Z-oriented parts.

Similar results were obtained for the storage modulus of Ultem samples having different thermal histories. Samples printed in the X and Y directions and cooled under different conditions were characterized by almost the same dependence of the storage modulus on the temperature. However, for samples printed in the Z direction and cooled in the printer, oven, and at room temperature, the reductions in the storage modulus in the glassy region were 13%, 108%, and 39%, respectfully, if compared with the results obtained for cooling in the printer. The glass transition temperature, a physical characteristic of Ultem material but not of a printed structure, was not affected by the printing direction (X, Y, Z) and was approx. 186 ± 4 °C.

Optical microscopy and SEM revealed that the void distribution in the transverse cross-sections of samples printed in the X and Y directions was the same, indicating a similar degree of fiber-to-fiber fusion, and completely different for samples printed in the Z direction, which displayed rough structures of extruded filaments. Different thermal histories did not significantly affect Ultem samples.

Thus, based on the results obtained for the tensile and thermophysical properties of Ultem printed parts, it can be concluded that removal from the printer and cooling in an oven with similar post-printing cooling conditions did not have significant consequences on these properties. Thus, it is possible to improve the productivity and efficiency of 3D printer machines by applying such a modified cooling procedure.

## Figures and Tables

**Figure 1 polymers-15-00324-f001:**
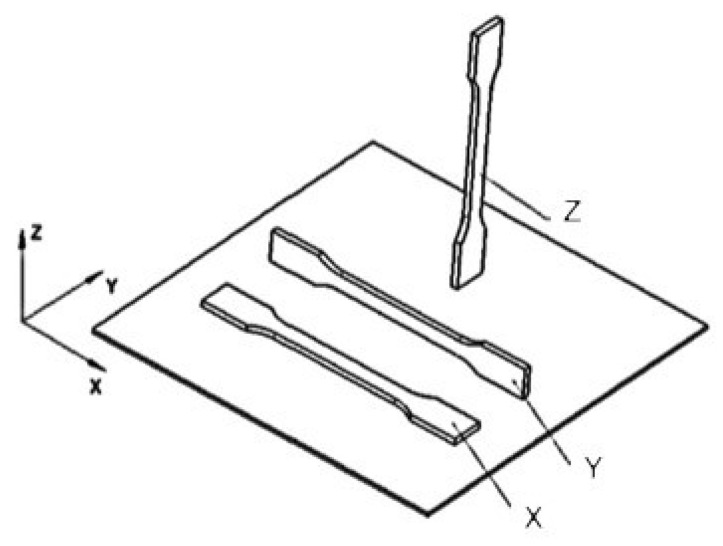
The orientation of manufactured tensile test specimens during the 3D printing process.

**Figure 2 polymers-15-00324-f002:**
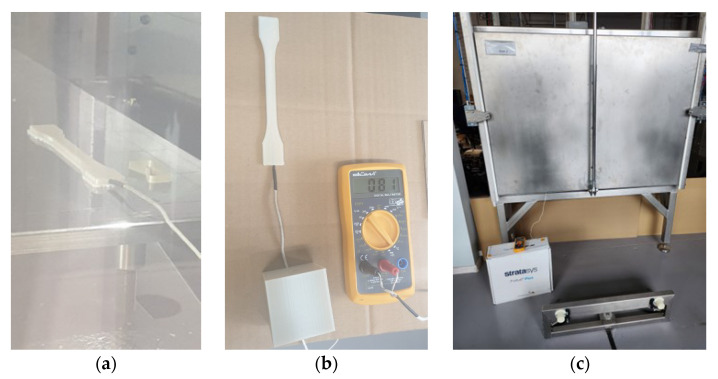
Ultem samples during post-printing cooling process in Stratasys F900 printer (**a**), at room temperature conditions (**b**), and in fully automatized oven with the sample inside it (**c**).

**Figure 3 polymers-15-00324-f003:**
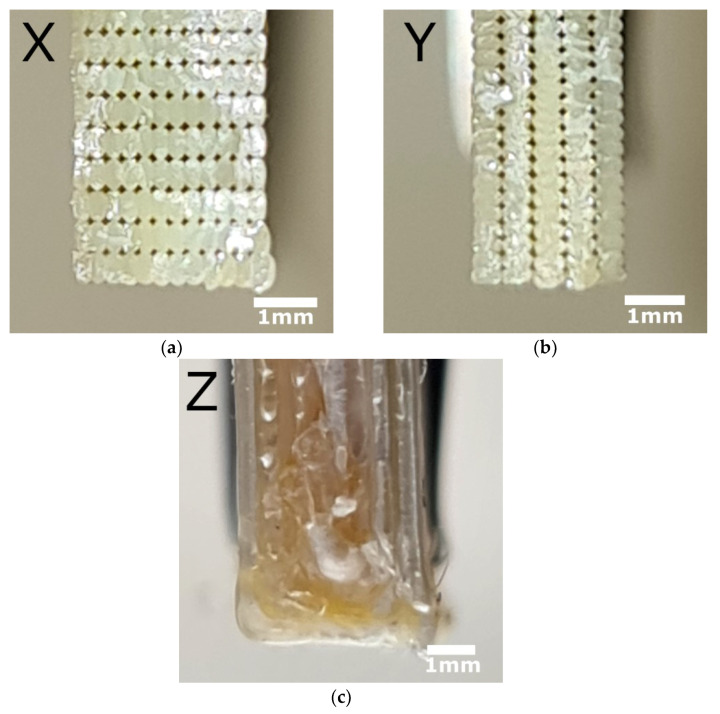
Transverse cross-sections of Ultem samples after cooling at RT printed in X (**a**), Y (**b**), and Z (**c**) directions and studied by optical microscopy.

**Figure 4 polymers-15-00324-f004:**
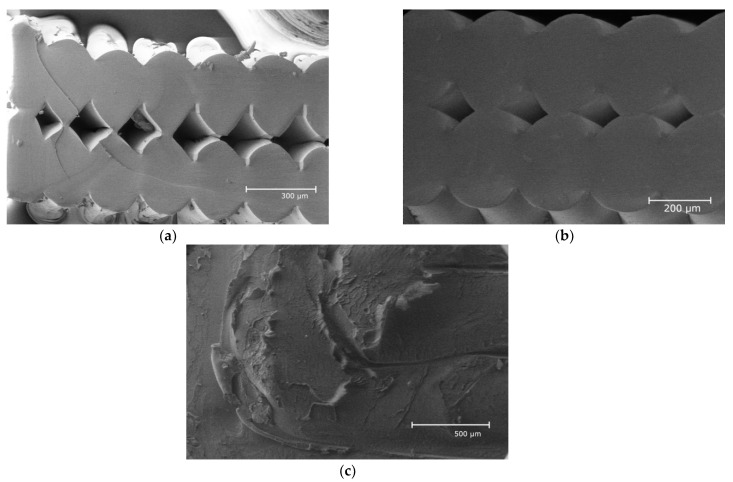
Transverse cross-sections of Ultem samples after cooling at RT printed in X (**a**), Y (**b**), and Z (**c**) directions and studied by SEM.

**Figure 5 polymers-15-00324-f005:**
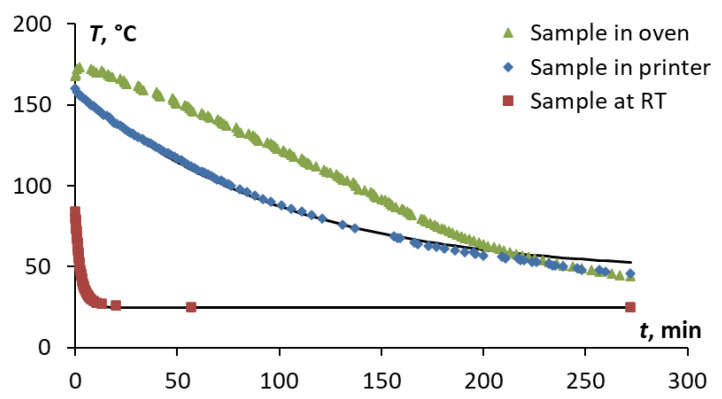
Temperature evolution in Ultem sample vs. time during the cooling process in the printer, oven, and at RT (as indicated on the graph). Symbols are experimental data, lines—evaluation by Equation (1).

**Figure 6 polymers-15-00324-f006:**
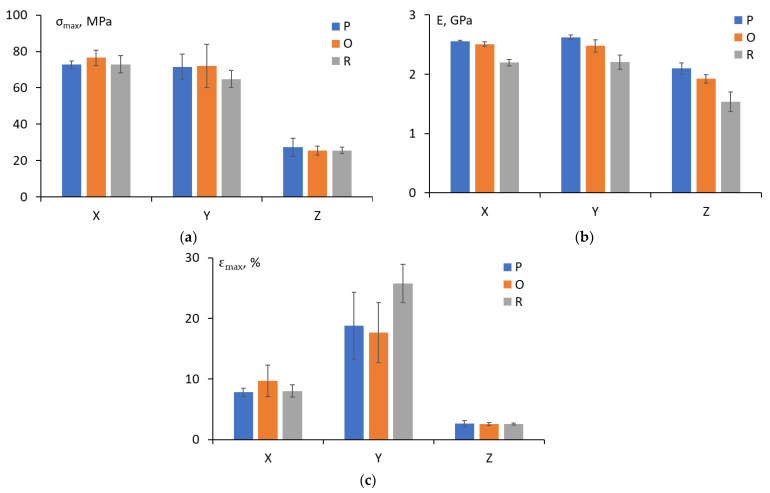
Tensile strength (**a**), elastic modulus (**b**), and maximal deformation (**c**) of Ultem in relation to printing direction (the cooling conditions are indicated on the graph as P, O, and R).

**Figure 7 polymers-15-00324-f007:**
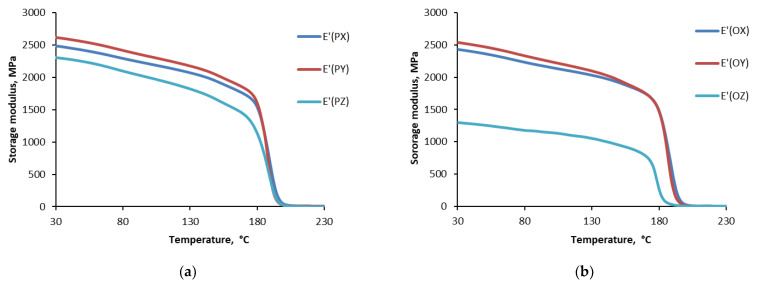
The storage modulus of ULTEM 9085 CG samples vs. temperature when cooled in the printer (**a**), oven (**b**), and at room temperature (**c**) and printed in X, Y, and Z directions.

**Figure 8 polymers-15-00324-f008:**
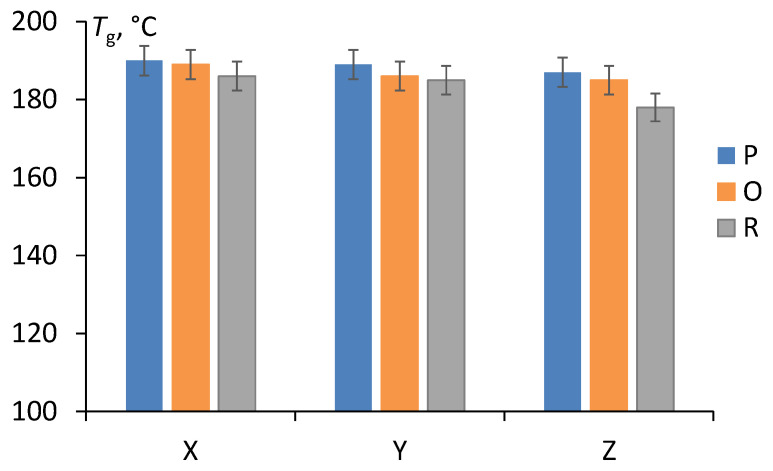
The glass transition temperature of Ultem samples cooled in the printer (P), oven (O), and at room temperature (R) and printed in X, Y and Z directions.

**Figure 9 polymers-15-00324-f009:**
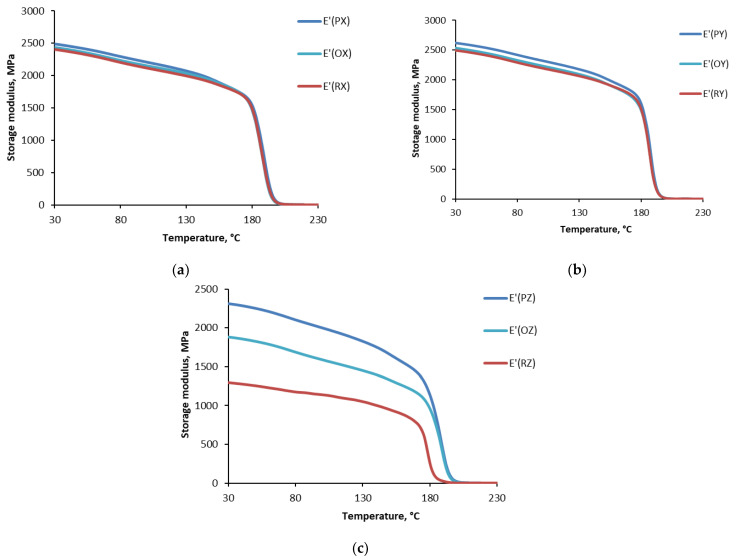
The storage modulus of Ultem samples vs. temperature printed in X (**a**), Y (**b**), and Z (**c**) directions. The cooling conditions (printer: P, oven: O, room temperature: R) are provided in the legend.

**Table 1 polymers-15-00324-t001:** Printing parameters.

Parameter	Value
Raster width	0.508 mm
Contour width	0 mm
Slice height	0.254 mm
Contour to raster air gap	0 mm
Raster to raster air gap	0 mm
Raster angle	0°
Infill density	100%

## Data Availability

The data presented in this study are available on request from the corresponding author.

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
