# Peer review of "Effect of Post-Printing Cooling Conditions on the Properties of ULTEM Printed Parts"

_polymers, 2023, doi:10.3390/polym15020324_

Round 1

Reviewer 1 Report

The aim of this paper is to estimate the effect of the post-printing cooling conditions on the tensile and physical properties of Ultem 9085 printed test specimens manufactured using FDM technology.

The study of the anisotropy of FDM printed parts is nowadays an interesting subject from a scientific and industrial point of view, however, unfortunately, the research developed by the authors presents certain limitations.

1.- The authors study several thermal post-processing conditions in test specimens fabricated in three main printing directions (X, Y and Z). The study mainly involves three boundary conditions: having the specimens 4 hours inside the printer or in an oven or immediately extracting the samples from the printer at room temperature after manufacturing. Unfortunately, the authors only take a single temperature of 45ºC for the evaluation of the influence of temperature, without detailing the reason why they have chosen this value. With a single temperature value, it is not possible to evaluate the influence of this parameter on the anisotropy of the printed specimens, specifically on the specific volume of voids in each printed direction.

Additionally, the authors do not detail how they achieve a uniform temperature of 45ºC inside the printer. In this case, there is a temperature gradient that the authors must characterize as boundary conditions. These boundary conditions affect different sections of the test specimens differently. It is recommended that the authors increase the number of tests including a greater number of temperatures and characterize the temperature boundary conditions inside the printer and on the different layers of the analyzed specimens.

2.- In section 3.4 morphology of the fracture surface, the authors do not refer to the test or standards used. Additionally, the authors do not include SEM pictures that justify the analysis of the fracture performed. Figure 8 does not show the results or type of fracture on the test specimens. Thus, it is not possible to evaluate in detail the results of the fracture obtained in front of the boundary conditions. Authors are recommended to increase the level of detail of this item by comparing fracture results obtained in the test specimens with SEM images, evaluating the different boundary conditions analyzed in the paper.

Reviewer 2 Report

Overall Comment:

The manuscript did not criticise in all aspects. For example: The printing nozzle moving in X- & Y- axis, is supposable that both printing axis is not affected in terms of time & quality, as agreed in the manuscript. However, it found slight differences in both printing axes, yet the authors are not discussing it. Suggest adding more critical discussion. 

The discussion is more likely a report of the results found, but not discussing. 

2. Materials & Methods

Suggest to adding the specimen's schematic in X-, Y- & Z- printing axis, this will aid readers in understanding the methodology in a view.  

The testing standard is missing

3. Results and Discussion

Figure 3 is not needed since it did not show any critical differences and numerical data were shown in figure 4.

When come to the comparison of different cooling methods, the critical part is the crystalline structure of the polymer, bonding status on Z-axis layering and perhaps internal stresses also. Yet the authors failed to discussion on these perspectives. Suggest revising the section. 

I am not quite sure what's the angle taken for the morphology figure. Suggest adding more explanation or even better explaining it in a schematic figure. 

Reviewer 3 Report

The main idea considered for publishing the paper is the presentation of the results of experimental studies concerned with post-printing cooling conditions on the tensile and thermophysical properties of ULTEM® 9085 3D printed with the use of the FDM technique.

My comments as a reviewer are as follow:

Remark 1

 The title is too long and has to be shortened. e.g “Effect of post-printing cooling conditions on mechanical and thermophysical properties of ULTEM® 9085 material” or something different.

Remark 2

I understand the main intention of the Authors that the application of post-processing cooling conditions with the use of an oven is reasonable because it allows to increase in the productibility of the 3D printer, nevertheless, it is rather obvious that the application of similar cooling conditions should not affect mechanical and thermophysical conditions. In my opinion, it is an attempt to do science from applied engineering solutions.

Remark 3

Line 66, page 2 “To the best of our knowledge, no similar research has been found in literature” I’m a very sceptic when I see a statement like this. It only means that the authors did not make a detailed literature survey. I spent 30 minutes, checking a few research papers databases and I found a few papers that are focused on a similar research problem. Some of them I placed below.

https://doi.org/10.1016/j.procs.2022.01.332

https://doi.org/10.1016/j.progpolymsci.2021.101411

https://doi.org/10.1016/j.promfg.2019.06.195

https://doi.org/10.1016/j.polymertesting.2018.05.020

A detailed literature survey is a crucial issue and always has to be done. If you don’t do it, you will define and try to solve problems that many researchers did in the past. In my opinion, the literature survey in that form is not acceptable.

Remark 4

Line 76, page 2. “These results will positively contribute to the optimization of the 3D printing technology of polymer parts”  I would recommend avoiding terms like optimisation. If you use it you have to define the optimisation function and criteria that were applied to obtain it. In my opinion, the proposed by the Authors solution allows for improving producibility and increases the efficiency of utilizing the 3D printer machine.

Remark 5

Figure 2, In my opinion, figure 2a is multiplied in figure 2b and does not provide new information. I would recommend deleting it.

Remark 6

I recommend adding to particular charts the small drawing illustrating the orientation of manufactured tensile test specimens during the 3D printing process. It will allow a better understanding of the applied conditions during the tensile test.

Remark 7

Page 6, Lines 194 to 198, this conclusion refers to typical properties of parts manufactured additively with the use of the FDM technique and it is not related to applied cooling post-processing. The formulated conclusion means that the Authors applied 3D printing parameters recommended by Stratasys Corp. and obtained similar results of mechanical properties. It only means that they properly utilised the Fortus M900 machine.

Remark 8

I’m curious about figure 4c. Could you add some additional comments on why the result obtained for the oven solution is better than for the solution where the cooling process in the 3D printer chamber was applied?

Remark 9

Page 8, lines 242 to 245. I can not agree with this conclusion. Does it mean that users of FDM machines should avoid building objects in 0Z orientation? It's impossible because the 3D printing process is typically carried out in the 0Z direction. Furthermore, the mechanical properties depend not only on the adopted parameters like cooling rate but also they are related to the layer thickness, nozzle and working chamber temperature value, filling orientations, etc. Taking into consideration the applied 3D printer it will be difficult to change it.  Stratasys is sceptical to modify these parameters. They are defined by them as optimal. In my opinion, this conclusion is rather not proper.

Remark 10

Figure 8,

For me, these photographs do not allow a state that the cooling rate somehow affects the mechanical conditions of tested material samples. Furthermore, it can be seen that the structure of the specimen is not homogenous so it strongly affects the mechanical properties. In my opinion, the Authors should use an optical microscope but with the use of larger magnification and also SEM to properly evaluate the fractography of the samples.

Remark 11

Conclusions are very oblique and do not provide a new scientific value. This section after improving the previous sections should be definitely rebuilt.

Remark 12

The number of references is very limited. There is a great number of very interesting papers related to studies of the Ultem material, FDM technique as well as mechanical characterisation of fabricated materials. Please do a literature study again.

Dear Authors,

With deep regret, I have to state that the submitted paper has a lot of drawbacks and in my opinion, it does not provide a new valuable scientific value. Nevertheless, I hope you will consider my remarks and improve the manuscript. Then it can be reconsidered for publishing. If you miss some of my comments I would recommend rejecting this paper.

Round 2

Reviewer 1 Report

The authors have addressed satisfactorily the points raised during the review.

Reviewer 2 Report

good

Reviewer 3 Report

Dear Authors,

I accept answers regarding my remarks. Nevertheless, I still have an impression that the scientific soundness of this paper is rather poor and it doesn't provide significant value to the field of FDM/FFF 3D printing technique as well as studies related to Ultem filament.